# Corilagin in Cancer: A Critical Evaluation of Anticancer Activities and Molecular Mechanisms

**DOI:** 10.3390/molecules24183399

**Published:** 2019-09-19

**Authors:** Ashutosh Gupta, Amit Kumar Singh, Ramesh Kumar, Risha Ganguly, Harvesh Kumar Rana, Prabhash Kumar Pandey, Gautam Sethi, Anupam Bishayee, Abhay K. Pandey

**Affiliations:** 1Department of Biochemistry, University of Allahabad, Allahabad 211 002, Uttar Pradesh, India; ashutosh8998@gmail.com (A.G.); amitfbs21@gmail.com (A.K.S.); rameshbiochem91@gmail.com (R.K.); rishaganguly53@gmail.com (R.G.); harshversity@gmail.com (H.K.R.); pandey.prabhash21@gmail.com (P.K.P.); 2Department of Pharmacology, Yong Loo Lin School of Medicine, National University of Singapore, Singapore 117600, Singapore; phcgs@nus.edu.sg; 3Lake Erie College of Osteopathic Medicine, Bradenton, FL 34211, USA

**Keywords:** Corilagin, bioavailability, anticancer activity, signaling pathways, safety evaluation

## Abstract

Corilagin (β-1-*O*-galloyl-3,6-(*R*)-hexahydroxydiphenoyl-d-glucose), an ellagitannin, is one of the major bioactive compounds present in various plants. Ellagitannins belong to the hydrolyzable tannins, a group of polyphenols. Corilagin shows broad-spectrum biological, and therapeutic activities, such as antioxidant, anti-inflammatory, hepatoprotective, and antitumor actions. Natural compounds possessing antitumor activities have attracted significant attention for treatment of cancer. Corilagin has shown inhibitory activity against the growth of numerous cancer cells by prompting cell cycle arrest at the G_2_/M phase and augmented apoptosis. Corilagin-induced apoptosis and autophagic cell death depends on production of intracellular reactive oxygen species in breast cancer cell line. It blocks the activation of both the canonical Smad and non-canonical extracellular-signal-regulated kinase/Akt (protein kinase B) pathways. The potential apoptotic action of corilagin is mediated by altered expression of procaspase-3, procaspase-8, procaspase-9, poly (ADP ribose) polymerase, and Bcl-2 Bax. In nude mice, corilagin suppressed cholangiocarcinoma growth and downregulated the expression of Notch1 and mammalian target of rapamycin. The aim of this review is to summarize the anticancer efficacy of corilagin with an emphasis on the molecular mechanisms involving various signaling pathways in tumor cells.

## 1. Introduction

Cancer is a leading non-communicable disorder and has been established as a serious health threat in developed and developing nations that differs in the age of onset, invasiveness, and the response towards treatment [1]. The worldwide cancer burden was assessed to have increased to 18.1 million new cases and 9.6 million deaths in 2018. Europe accounts for nearly 23.4% of the global cancer incidence and 20.3% of the cancer mortality. The Americas account for 21.0% of cases and 14.4% of deaths globally. As compared to other regions of the world, the ratio of mortality from cancer in Asia and Africa (57.3% and 7.3%, respectively) are greater than the magnitudes of cancer incidence (48.4% and 5.8%, respectively), because of higher frequency of certain types of cancer in these regions, which are linked with poorer prognosis and higher death rates, along with limited resources to timely diagnosis and treatment in several countries [2]. Lung cancer is reported to cause maximum deaths (18.4% of the total death due to cancer), because of the poor prognosis of lung cancer throughout the world, followed by colorectal (9.2%), stomach (8.2%), and liver (8.2%) cancers. Breast cancer is the fifth major cause of death (6.6%), which could be due to the favorable prognosis, at least in more developed countries [2,3]. Cancer death is predicted to increase to 20.3 million by the year 2026 because of change in size and arrangement of the population. Among the female population, cervix, breast, and ovarian cancers accounts for 34% of all deaths reported due to cancer.

Traditional and recently emerging methods, such as chemotherapy, radiotherapy, catalytic therapy, and photodynamic therapy, have not succeeded in countering the consequence of cancer to any drastic extent, and this limitation has directed researchers to explore alternative treatment options [4]. The recognition of molecules in providing protection against cancer without adverse effects remains the most important objective. Bioactive phytochemicals derived from food and medicinal plant sources have contributed immensely in the drug discovery process as an alternative source that can be utilized as curative agents against a diversity of ailments, including cancer [5,6]. Phytochemicals such as polyphenols, alkaloids, tannins, flavonoids, terpenes, taxanes (diterpenes), saponins, vitamins, lignans, glycosides, oils, gums, and other metabolites play noteworthy roles in constraining cancer cells by altering proteins, enzymes, and signaling pathways [7,8]. About 60% of newly discovered anticancer drugs currently known are derived from natural products [9,10]. Chemotherapeutic agents of natural origin used currently include paclitaxel (taxol) and its semisynthetic analogue docetaxel (taxotere) acquired from the Pacific yew (*Taxus brevifolia)* along with vinblastine and vincristine isolated from the periwinkle *(Vinca rosea*) [11,12,13].

Tannins are high molecular weight phenolic compounds, ubiquitously present in bark, stem, root, and fruit of plants, and are soluble in alcohol and water [14]. Tannins and associated compounds, such as ellagitannin, gallotannin, and complex tannins, have shown cytotoxic activities against various human cancer cell lines, including epidermoid carcinoma, lung carcinoma, ileocecal adenocarcinoma, medulloblastoma, and malignant melanoma cells [15]. Ellagitannins are hydrolysable tannins that exert numerous therapeutically beneficial activities. The antiproliferative, anticarcinogenic, and chemopreventive activities of ellagitannins have drawn increasing attention in recent years [16]. Cuphiin D1, a macrocyclic hydrolysable tannin, exhibited cytotoxicity against HL-60 cells [17]. Maplexins, the gallotannins present in the red maple (*Acer rubrum*), have exhibited potent anticancer activity against breast (MCF-7) and colon (HCT-116) cancer cells [18]. Other tannins, namely, chebulinic acid and chebulagic acid (isolated from *Geranium wilfordii* Maxim), punicalagin, and ellagic acid (isolated from *Punica granatum*), have also shown antitumor activity against various cell lines in vitro by inducing apoptosis [19,20].

Corilagin (β-1-*O*-galloyl-3,6-(*R*)-hexahydroxydiphenoyl-d-glucose) is a natural ellagitannin (ET) found in a wide range of plants. In recent years, corilagin has been documented to possess many biological and pharmacological attributes, such as antioxidant [21], anti-inflammatory [22], hepatoprotective [23], antimicrobial [24], antihypertensive [25], antidiabetic [21,26], and antitumor activities [27,28,29,30]. It inhibited reverse transcriptase activity of RNA tumor viruses and possessed significant antistaphylococcal activity (MIC 25 μg/mL) [31]. The antiproliferative property of ET has been attributed to multiple mechanisms, comprising cell cycle inhibition, apoptosis via the mitochondrial pathway, and self-destruction after replication [32]. The chemopreventive actions of ET and derived compounds have also been correlated with their antioxidant potential that changes with the number of hydroxyl groups [33]. Since corilagin has received substantial attention for its versatile medicinal activities, a systematic review regarding its therapeutic attributes, particularly as antitumor agents, is desired. Recently, Li and co-workers described the pharmacological role of corilagin in countering various diseases, along with an overview of limited cancer studies [34]. In the current review, we discuss the efficacy of corilagin against various types of cancer with a focus on the detailed molecular mechanisms involving various signaling pathways of tumor cells.

## 2. Methodology for Literature Review

Numerous databases, including PubMed, Scopus, EBSCOhost, and ScienceDirect, were used for exploration and collection of literature. The Preferred Reporting Items for Systematic Reviews and Meta-Analysis (PRISMA) criteria [35], suggested for writing systematic reviews was followed. The articles written in English language only were incorporated in the current review. Major keywords, such as corilagin, cancer, signaling pathway, pharmacological effect, prevention, treatment, bioavailability, and in vivo and in vitro studies, were utilized during literature search.

## 3. Distribution of Corilagin and Its Physicochemical Properties

Corilagin (C_27_H_22_O_18_, molecular weight 634) (Figure 1) is an ellagitannin belonging to the polyphenol family. Physically, it is an off-white acicular crystalline powder soluble in dimethyl sulfoxide, methyl alcohol, acetone, and ethyl alcohol. It is present in various plants species (Table 1) and has diverse therapeutic properties. Some of the major families well-known for the presence of corilagin include Combretaceae *(Terminalia catappa* L.), Euphorbiaceae (*Euphorbia longana* Lam., *Phyllanthus emblica* L., *P. urinaria* L., *P. tenellus* Roxb., *P. niruri* L., and *Acalypha australis* L.), Geraniaceae (*Geranium sibiricum* L.), Polygonaceae (*Polygonum chinense* L.), and Saururaceae (*Saururus chinensis* (Lour.) Bail.) [34,36,37,38,39,40]. Corilagin is mainly isolated from the aerial parts, such as leaf, stem, flower, fruits, and seeds, of various plants.

## 4. Bioavailability of Corilagin

Bioavailability of ellagitannins is low in human and animal models due to their hydrophobic nature. Previous studies have shown that corilagin is hydrolyzed to ellagic acid and gallic acid under physiological conditions in the intestine, which are moderately absorbed and metabolized by gut microbiota [34]. Recently, HPLC-Q-TOFMS/MS has been used to characterize corilagin and its metabolites in various biological samples. The study reported the presence of corilagin and their metabolites in plasma and liver tissue [41]. In vivo studies in rat and mice after oral administration of corilagin (1500 mg/kg) were conducted by Reddy et al. [42]. They analyzed plasma samples at different time intervals using HPLC–ESI–MS, which showed peak bioavailability of corilagin at 2 h with maximum concentration of about 55 µg/mL in blood, and half-life was found to be about 6 h [42]. In another study, Sprague–Dawley rats were fed with a single dose of corilagin (50 mg/kg) orally. Blood analysis at various time intervals spanning over 24 h was conducted using UPLC coupled with the photodiode array method. The results revealed the maximum bioavailability of corilagin in plasma was 1.8 µg/mL after 2 h of oral dosing [43].

Hydrolytic products of corilagin (e.g., ellagic acid and gallic acid) exhibited differences in bioavailability and half-life as compared to the parent compound in plasma. Hence, pharmacological action of corilagin is the result of cumulative activity of corilagin as well as ellagic acid and gallic acid. Moreover, ellagic acid and gallic acid exhibited anticancer activity by inducing apoptosis, downregulating genes involved in cell cycle and angiogenesis, and stimulating a cellular immune response [44,45]. To overcome bioavailability issues, new delivery systems are continuously being developed to promote the bioavailability of the compounds into the systemic circulation by using liposomes, nanoparticles, microemulsions, and polymeric implants [44].

## 5. Anticancer Activity of Corilagin

Research to reveal the anticancer potential of corilagin was started in 1985 [88]. In RNA tumor virus, it inhibited reverse transcriptase activity [89]. Studies on in vivo growth inhibition activity of corilagin against sarcoma-180, a murine fibrosarcoma cell line, suggested the improvement in life span of mice treated with corilagin by 36.1% [90]. Other researchers also observed the suppression of cancer cell growth by inhibiting the DNA relaxation mediated by topoisomerase-I [91,92,93]. It was reported that corilagin suppressed the discharge of tumor necrosis factor-α (TNF-α) from carcinoma cells. It was suggested that TNF-α stimulated the proliferation and advancement of malignant cells during early stages of cancer development [94]. The effects of corilagin on various tumor cells and basic antitumor mechanisms are summarized in Table 2 and are discussed below.

### 5.1. Breast Cancer

Breast cancer is a primary cancer extensively affecting female health, and frequency and severity is rising throughout the world [95,96]. Other than heredity, the cumulative risk aspects are chiefly due to pregnancies at late age and little or no breastfeeding. Obesity or inactivity triggered by sedentary lifestyle can also increases the risk of breast cancer. However, appreciable advancement has been made in breast cancer treatment with the help of various therapies, viz., radiotherapy, chemotherapy, and endocrine therapy [97,98]. Several complications, such as drug resistance, tumor recurrence, or metastasis still persist following completion of medication. Therefore, it is essential to improve the techniques and further develop target specific drugs for the therapy of breast cancer. Tong and coworkers [28] reported that corilagin stimulated both autophagy and reactive oxygen species (ROS)-mediated apoptosis in breast cancer cell lines (MDA-MB-231 and MCF-7). Corilagin significantly increased ROS production in MCF-7 cells, while in combination with N-acetyl-L-cysteine, the opposite effect was observed exhibiting remarkable suppression of ROS production resulting in inhibition of cell death. This research indicated that corilagin could prevent the growth of breast cancer via ROS-mediated apoptosis and autophagy. Hence, corilagin may be considered as a new promising antitumor drug candidate for treating breast cancer [28]. In SK-BR3 cells, corilagin has been shown to enhance the level of receptor-interacting protein kinase-3 (RIP3) expression. RIP3 is a crucial component of the cellular system that causes the programmed necrotic cell death [99,100,101].

### 5.2. Cholangiocarcinoma

Cholangiocarcinoma (CCA) is a form of malignancy associated with the biliary tract. CCA has a higher mortality rate due to the challenging process of diagnosis [102]. The main cause of increasing CCA incidences could be associated with trade-offs between environmental and genetic factors [103]. The presently used chemotherapeutic drugs, including mitomycin and oxaliplatin, are vulnerable to resistance by tumor cells, hence compromising the drug efficacy [104,105]. Resection with the help of surgery is the only method of CCA treatment. Yue and coworkers [29] explored the potential anticancer effect of corilagin against CCA cell line phenotypes and suggested that it efficiently prevented cell multiplication and cell cycle progression, promoted apoptosis, and suppressed CCA cell metastasis (migration and invasion). Corilagin treatment could block CCA cell multiplication by introducing G_2_/M phase arrest, while in the case of MZ-Cha-1 cells, no remarkable alterations have been reported. Corilagin also repressed the mRNA level of Bcl-2 and enhanced caspase-3-mediated apoptotic gene expression in CCA cells [29].

Corilagin also inhibited Notch1 expression in CCA cells during the notch signaling pathway and thereby reduced the notch intracellular domain (NICD) form. In the absence of NICD, transcription factors (Lag1/Su(H)/CBF1) of the CSL family led to suppression of the transcription state, which controlled the action of Akt/phosphoinositide 3-kinase (PI3K) expression. In addition, *N*-(*N*-(3,5-difluorophenacetyl)-l-alanyl)-sphenylglycine *t*-butyl ester (DAPT, an inhibitor of the notch signal pathway) possibly impeded the expression of p-Akt, Notch1, and p-Erk1/2 protein. A blended treatment of DAPT with corilagin produced better antiproliferative activity. In NICD plasmid transfected CCA cells, the p-Erk1/2 and p-Akt expression were upregulated in comparison to control cells, which indicated that NICD directly affects the manifestation of downstream target genes, even though exogenous NICD resulted in diminished expression of Notch1, which acts as a negative feedback inhibitor [106]. These observations suggest that corilagin alters the manifestation of p-Erk1/2 and p-Akt, mammalian target of rapamycin (mTOR), and Notch1 proteins to control the occurrence and advancement of CCA by targeting the notch signaling pathway. Gu and coworkers [107] studied the anticancer potential of corilagin against CCA cell lines and reported that it inhibited proliferation and cell cycle progression, suppressed invasion and migration, and promoted CCA cell apoptosis. Moreover, corilagin downregulated the mRNA level of Bcl-2 and enhanced caspase-3.

### 5.3. Esophageal Cancer

Esophageal squamous cell carcinoma (ESCC) is a widespread intrusive esophageal cancer with poor detection. Corilagin effectively inhibited ESCC cell proliferation and induced apoptosis [108]. The effects were also validated in vivo using a xenograft mouse model. It caused noteworthy DNA damage in ESCC cells. Moreover, corilagin acted on the ubiquitin–proteasome pathway and diminished the expression of E3 ubiquitin ligase RING finger protein 8 (RNF8), consequently disabling the DNA damage repair response and ultimately triggering apoptosis. Further, corilagin in combination with cisplatin considerably increased the anticancer chemotherapeutic efficacy of cisplatin in vitro and in vivo [108]. Hence, corilagin might act as an adjunctive treatment to conventional chemotherapy in ESCC patients.

### 5.4. Gastric Cancer

In developing countries, gastric cancer is the third most commonly detected cancer in men and the major cause of cancer-related death [109,110]. Moreover, gastric cancer cells exhibit higher incidence of drug resistance towards commonly used chemotherapeutic drugs [111,112]. Usually for the management of gastric cancer, surgery is a prime treatment option but survival rate for even five years is low. The majority of patients relapse following surgery [113]. Xu and coworkers [114] studied the efficacy of corilagin on human gastric cancer cell lines (SGC7901 and BGC823) and suggested that it significantly inhibited cell proliferation, apoptosis, and autophagy. It was observed that corilagin fragmented poly (ADP ribose) polymerase (PARP, an indicator of caspase activation) and decreased the level of procaspase-8, procaspase-9, and procaspase-3 in BGC823 and SGC7901cells in a dose-dependent way, as detected by Annexin V/PI staining. Corilagin induced autophagy and ROS build-up in human gastric cancer cells, which is crucial for restricting the growth of cancer cells. Thus, it is likely that ROS build-up stimulated apoptosis of the corilagin-treated gastric cancer cells. Moreover, corilagin has been confirmed to exert a less toxic effect on normal human gastric mucosal epithelial cells (GES-1). Hence, corilagin could be a propitious agent for the cure and management of human gastric cancer [114].

### 5.5. Hepatocellular Carcinoma

Hepatocellular carcinoma (HCC) is the deadliest cancer throughout the world. The last few decades have shown its rapid rise throughout the world [115,116]. The HCC rates are frequently altered due to higher incidence of chronic hepatitis B virus (HBV) [117]. The preponderance of HCC patients usually exists in progressive and irredeemable stages and are not appropriate for surgery due to a narrow hepatic approach. Despite establishment of numerous non-surgical methods [118], surgical procedures have undergone much advancement [119]. However, none of these methods have been considerably useful in alleviation of patients from HCC. Ming et al. [120] conducted an in vivo xenograft experiment and reported that corilagin can prevent the growth of MHCC97-H tumor cells in a concentration-related manner and found to be more effective as compared to a cyclophosphamide (25 mg/kg)-treated group. Furthermore, corilagin reduced the expression of p-Akt and amplified p-p53 expression. P-p53, as well as p-Akt played critical roles in apoptosis or cell cycle inhibition. P-Akt could enhance p21^Cip1^ and p53 ubiquitylation and degradation, which act as inhibitors of cdc2/cyclin B1 proteins via the initiation of murine double minute 2 (Mdm2) and proliferating cell nuclear antigen (PCNA) [121,122]. Deng et al. [123] observed that corilagin downregulated the expression of p-Akt while it upregulated p53 protein expression in the SMMC-7721 cell line in a concentration-dependent manner. Break down of caspase-9, caspase-3, and PARP was also detected, which supported the stimulation of the intrinsic apoptotic pathway. Besides, caspase-8 is a vital protein of the extrinsic apoptotic pathways. Corilagin upregulated Fas and FasL signaling pathways, which in turn activated caspase-8 and thus triggered the extrinsic apoptosis pathway [123].

### 5.6. Lung Cancer

Recently, Bai et al. [30] reported the cytotoxic effect of longan (*Dimocarpus longan* Lour.) pericarp extract against the A549 lung cancer cell line in vitro. HPLC investigation suggested that longan possesses corilagin as one its major bioactive compound. On comparison of the activity of extracts with corilagin, it was found that corilagin exhibited an appreciable growth inhibitory effect on A549 cells with an IC_50_ value of 28.8 ± 1.2 µM. The propidium iodide staining further confirmed the morphological changes in the A549 cell nucleus.

### 5.7. Neural Cancer

Glioblastoma multiforme (GBM) is a lethal malignant tumor accountable for 42% of the total central nervous system tumors [124]. Currently, there is lack of appropriate treatment options for this malignancy. However, temozolomide (TMZ) is the most commonly recommended drug. TMZ is an alkylating agent, used for the treatment of this condition, but did not show satisfactory response, and it only caused an increase of life expectancy of the treated patients [125]. Hence, new drugs are immediately required for authentication and conceivable engagement in therapeutic protocols for antiglioma medications [126]. Furthermore, over the time a high number of gliomas become resistant to TMZ [127], emphasizing the need for discovery of new drugs to improve the treatment of TMZ-resistant tumors. Milani and coworkers [128] revealed that corilagin activated cell growth suppressive and pro-apoptotic activities in human glioma cell lines (T98G and U251) with stimulation of the apoptotic pathway. Yang et al. [129] observed that corilagin inhibited U251 cells at the G_2_/M stage of mitotic division while the U251-stem-like cells were blocked at the S phase. Corilagin stimulated the expression and inhibited the degradation of inhibitor of κBα (IκBα), blocked nuclear factor-κB (NF-κB) activation, suppressed the stimulated p65 protein moving into the nucleus, and thus inhibited the NF-κB signaling pathway and promoted apoptosis in neural tumor cells [129].

### 5.8. Ovarian Cancer

Ovarian cancer is the most prevalent type of gynecological cancer in females and the 5^th^ major cause of death [130]. The current anticancer therapy consists of a combination of platinum with paclitaxel, which may lead to severe toxicity. Hence, researchers and pharmaceutical companies are trying to develop new drugs for ovarian cancer that are effective with the least side effects. Ovarian cancer arises from the ovarian surface epithelium (OSE). During the normal ovulatory cycle, TGF-β prevents human OSE propagation and promotes apoptotic processes that may inhibit the surplus production of cells [131]. Furthermore, TGF-β can act as an inhibitor by preventing cell propagation in the initial phase of tumor progression [132,133]. TGF-β also triggers signaling pathways not associated with Smad pathways, such as those facilitated by members of the mitogen-activated protein kinase (MAPK) family (e.g., extracellular signal-regulated kinase, TAK1, c-Jun-NH_2_-kinase, and p38), and phosphatidylinositol 3-kinase [134]. Corilagin prevented the TGF-β secretion and also blocked the TGF-β-associated signaling proteins, such as p-Akt, p-Erk, and p-Smads. Jia and coworkers [39] studied the anticancer potential of corilagin against regular OSE cells (OSE-01, -02, and -03) and cancer cell lines (HO8910PM, SKOv3ip, and Hey) and suggested that corilagin arrested the G_2_/M phase of the cell cycle, possibly by downregulation of cyclin B1 and cdc2 through Myt1 and Wee1 regulation, and promoted apoptosis in ovarian cancer cells, whereas it was least cytotoxic to normal OSE cells. In SKOv3ip and Hey cells, corilagin decreased cyclin B1 and p-cdc2 (Tyr15) levels, which might be the molecular mechanism associated with cell cycle arrest. Consequently, epidermal growth factor (EGF), after stimulation by corilagin, suppressed the expression of Myt1 and p-Akt (Ser 473) proteins in SKOv3ip and Hey cells, suggesting that Akt/Myt1 inhibition also favors cell cycle arrest. Reverse phase protein array analysis indicated that in HO8910PM cells many signaling pathways were downregulated following treatment with corilagin, while western blot analysis verified these activities in the HO8910PM, Hey, and SKOv3ip cell lines [40].

## 6. Effect of Corilagin on Various Signaling Pathways of Cancer Cells

In normal cells, signaling pathways regulate various physiological processes and other essential functions. When such signals and mechanisms are challenged by various internal and external stimuli and avert cells to perform apoptosis, then normal cells undergo transformation to cancerous cells. A vast range of studies have revealed that interference with these transformed signals or mechanisms may contribute to anticancer effects. The following section summarizes various oncogenic signaling pathways, which are the possible targets of corilagin in neoplastic cells.

### 6.1. NF-κB Signaling Pathway

The pro-oncogenic NF-κB is a critical transcription factor that encompasses closely associated proteins. NF-κB usually exists in dimer form and attaches to a conserved sequence of DNA within the promoter region of target genes, known as the κB-B site. This upregulates target gene transcription by employing corepressors and coactivators [135]. The NF-κB pathway performs an essential function in oncogenesis by trans-activation of genes participating in apoptosis, proliferation, migration, invasion, metastasis, and angiogenesis [136]. Lee et al. [137] suggested that enhanced NF-κB expression intensely relates with accelerated advancement of tumor and reduced patient survival rates. Blocking NF-κB expression could trigger tumor cell apoptosis [138]. The family of NF-κB1 transcription factors consists of five members, namely RelA (p65), RelB, c-Rel, NF-κB1 (p50), and NF-κB2 (p52). All these members have a common Rel homology domain at the N-terminus that is essential for DNA binding and homo/hetero dimerization with the help of ankyrin repeats, holding the nuclear localization sequence of NF-κB [139]. Corilagin potentially prevented the phosphorylation of RelA (p65) in PC12 cells in a concentration-related manner. At 1–10 µM concentration, nearly complete inhibition of Rel A (p65) was observed. The expression of phospho-IκBα by Aβ25-35 depends on the level of Rel A (p65) phosphorylation, which is appreciably lowered by corilagin. These results clearly suggested that reduction of NF-κB activation by corilagin possibly led to attenuation of prostaglandin E2 (PGE2), cyclooxygenase-2 (COX-2), nitric oxide (NO), einducible nitric oxide synthase (iNOS), and TNF-α in PC-12 cells in pheochromocytoma, a hormone secreting tumor of the adrenal medulla (Figure 2). Western blot analysis showed that a higher amount of corilagin increased the cytosolic expression of IKBα protein, while it decreased the nuclear expression of NF-κB/p65 protein [129]. In addition, the NF-κB signaling pathway is closely related with the stimulation of MAP kinases, which trigger downstream transcription factors that upregulate expression of inflammatory genes [140]. These signaling pathways are thoroughly recognized in mammalian systems and comprise p38 MAP kinase (p38), c-Jun N-terminal kinase (JNK), and mitogenic signaling extracellular signal-regulated kinase 1/2 (Erk1/2) [141].

### 6.2. Notch-mTOR Signaling Pathway

The notch signaling pathway is a progressively conserved pathway that plays a critical role in embryonic and postpartum advancement in various organisms [142]. It is a well-organized, multi-tiered, highly regulated cascade of signaling events. Both the ligands and the receptors are transmembrane proteins, and the pathway is activated when the ligands make association with their respective receptors [143]. Due to multi-dimensional behavior, this pathway is prone towards unusual activation of signaling components and is related to numerous abnormalities, such as various developmental syndromes and carcinogenicity [144]. In one study, western blot analysis revealed that corilagin exhibited a remarkable effect on Notch1 at certain time gaps (i.e., 12, 24, and 48 h) and potentially suppressed the expression of mTOR at 48 h. The results also indicated that corilagin treatment expressively reduced p-Erk1/2 and p-Akt expression at 48 h. Co-immunoprecripitation assay showed the interactions between mastermind like protein (MAML1), NICD, and P300 by co-transfection of CCA cells with HA-NICD, MYC-NICD, HA-P300, and Flag-MAML1. The study also indicated that co-immunoprecipitation occurred between HA-P300/ MYC-NICD and Flag-MAML1/HA-NICD, which suggested that in CCA cells there was interaction among NICD with P300 and MAML1. These findings support that corilagin alters the expression of downstream target proteins namely, Notch1, mTOR, p-Erk1/2, and p-Akt, to control the initiation and growth of CCA via regulation of the notch signaling pathway (Figure 3) [108].

### 6.3. TGF-β Signaling Pathway

The TGF-β pathway regulates vital functions of cells, e.g., extracellular matrix production, cell growth, differentiation, invasion, angiogenesis, apoptosis, motility, and immune response [145]. In the case of cancer, TGF-β signaling plays dual roles: during the primary phase of cancer progression it acts as a tumor suppressor, whereas in the late phase, it promotes tumor advancement, facilitating metastasis and invasion by modifying the immune response and microenvironment of the tumor. Normal cells synthesize all the three TGF-β isoforms (e.g., TGF-β1, -β2, and -β3) as precursor molecules, which consist of a pro-peptide region in addition to the TGF-β homodimer [146]. In cancer cells, various isoforms of TGF-βs induce secretion of pro-matrix metalloproteinase (MMP), loss of contact between cells, upregulation of N-cadherin, downregulation of E-cadherin, and attainment of a fibroblastoid phenotype. All these processes are in conformity with an epithelial–mesenchymal transition [147,148,149]. Moreover, TGF-β makes association with type I (ThRI) and type II (ThRII) receptors. After attachment of ThRII with ligands, activation of ThRI takes place, which triggers phosphorylation of receptor-regulated Smads. The phosphorylated Smads, in turn, bind to the co-Smad and Smad4. This complex further moves to the nucleus causing gene expression alterations. TGF-β also triggers Smad-independent pathways, comprising those facilitated by the PI3K and MAPK family members (e.g., c-Jun-NH_2_-kinase, TAK1, p38, and extracellular signal-regulated kinase) [134]. Jia and coworkers [39] suggested that corilagin could prevent the phosphorylation of Smad2, Akt, and Erk in ovarian tumor cells by downregulation of TGF-β that ultimately led to the cancer cell apoptosis [39]. Cell cycle arrest by corilagin in S- and G_2_/M phases might be mediated by inhibition of the cyclin B1/cdc2 complex, which accomplishes an essential role in regulating G_2_ to M phase progression in SKOv3ip cells (Figure 4) [150,151].

## 7. Safety Evaluation of Corilagin

Corilagin did not produce any harmful effects on the viability of the murine macrophage cell line (RAW264.7) in in vitro assay, which suggested that corilagin is non-toxic in cell culture [34]. Another study by Reddy and co-workers [42] further corroborated that corilagin treatment (up to 1000 µM) did not induce any toxicity on Huh7 hepatoma cells during an in vitro cell viability assay. In addition, corilagin was found to be safe with no evidence of its mutagenic activity based on the Ames test [152]. Use of corilagin has also been shown to be safe during in vivo studies as it does not produce acute and sub-acute toxic effects. An in vivo acute toxicity study in BALB/c mice revealed that corilagin exhibited maximum tolerance level up to 3500 mg/kg (LD_50_ 3500–5000 mg/kg). The findings suggested that it can be safe and non-toxic even at higher dosages. Moreover, during a sub-acute toxicity study, the daily oral dose of corilagin (1000 mg/kg) for four weeks did not show any adverse effect on body weight and behavior of the mice [42]. Therefore, the above mentioned studies indicated that corilagin was almost safe without any adverse effects and virtually nontoxic to normal cells or tissues.

## 8. Limitations and Future Prospects

Corilagin, a natural dietary agent, has the potential for preventing and treating several types of cancer. It modulates numerous signaling pathways for its anticancer action. Since the number of in vivo studies is still less, the findings on the cancer prevention and treatment needs further investigation. A recent study identified the presence of corilagin and its twenty four metabolites in rat blood and tissues after oral administration. So far, biological activity of only two metabolites, i.e., ellagic acid and gallic acid, has been studied properly [41]. Therefore, it is important to investigate the pharmacological activities and the mechanism of action of metabolites for better understanding of corilagin as an anticancer agent. Furthermore, the low bioavailability of corilagin and its metabolites is a major concern. To overcome this issue, there is need to explore existing and newer drug delivery systems as viable options for delivering remedial concentrations of corilagin into the systemic circulation.

## 9. Conclusions

Corilagin is one of the essential phytoconstituents of various plants of ethnopharmacological interest and a promising versatile herbal medicinal agent. It prevents the growth of various types of cancer cells in vitro and CCA cells in vivo. Most of the studies have been performed in vitro with only few in vivo results and possibly no clinical study has been reported. Corilagin not only exhibited encouraging antitumor activities, but it also showed the least toxicity towards normal cells and tissues. Therefore, corilagin is a potent and promising anticancer drug candidate. The major anticancer effects of corilagin are brought about by inhibition of cell proliferation, induction of apoptosis, and cell cycle arrest. Corilagin modulates NF-κB, Notch-mTOR, and TGF-β signaling pathways. The maximum bioavailability in rats has been shown to be very low after 2 h of oral administration, which is a major concern. Still, the mechanisms underlying the antitumor potential of corilagin are not completely understood and more in vivo studies are required. Further clinical studies on anticancer efficacy of corilagin are also needed to investigate the systemic bioavailability of corilagin from dietary sources or specific drug formulations, optimum dose, and duration.

## Figures and Tables

**Figure 1 molecules-24-03399-f001:**
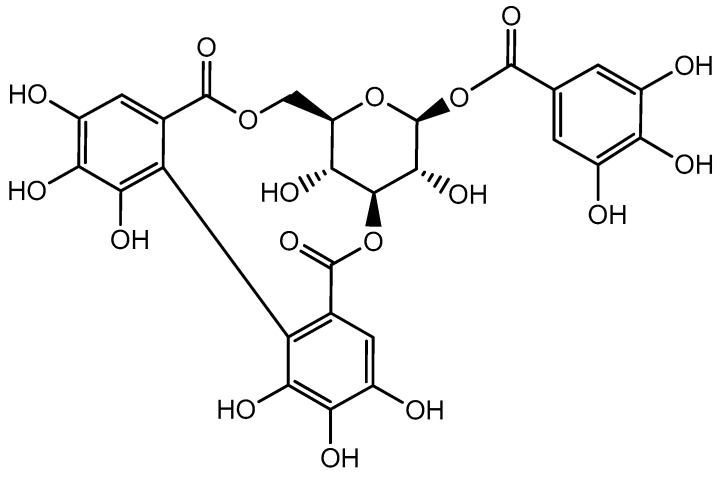
The chemical structure of corilagin.

**Figure 2 molecules-24-03399-f002:**
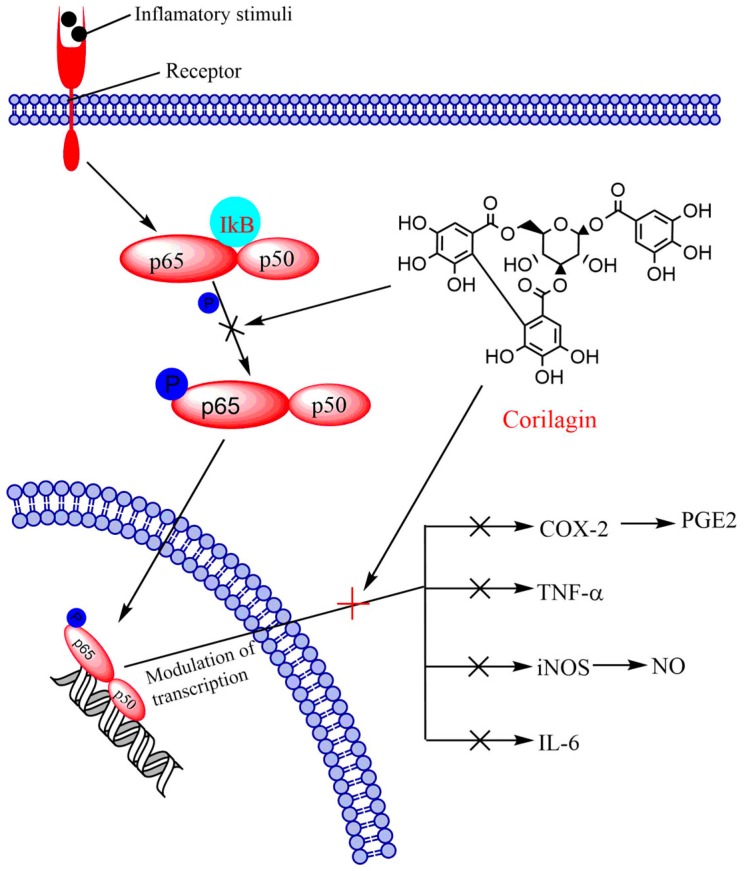
The effect of corilagin on NF-κB and related pathway in neural cancer.

**Figure 3 molecules-24-03399-f003:**
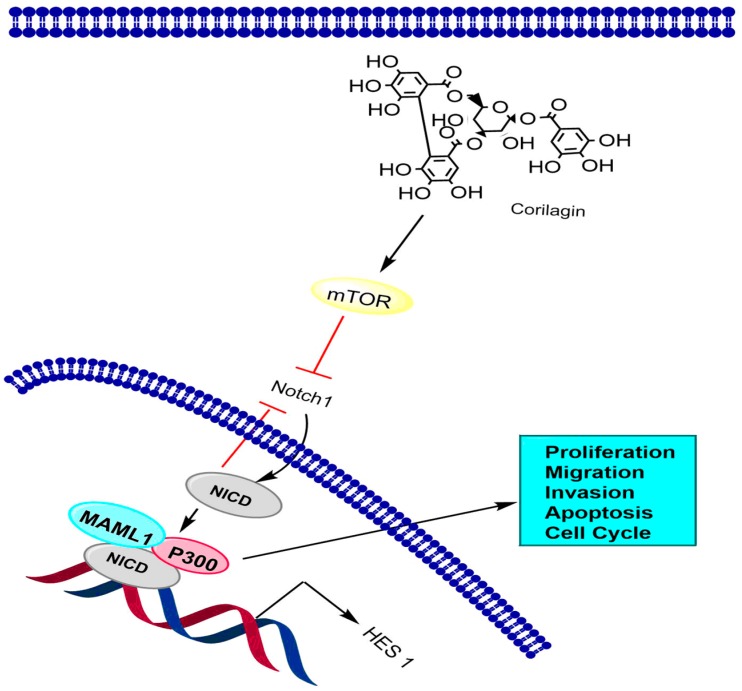
The effect of corilagin on the Notch-mTOR signaling pathway in cholangiocarcinoma. NICD is notch intracellular domain.

**Figure 4 molecules-24-03399-f004:**
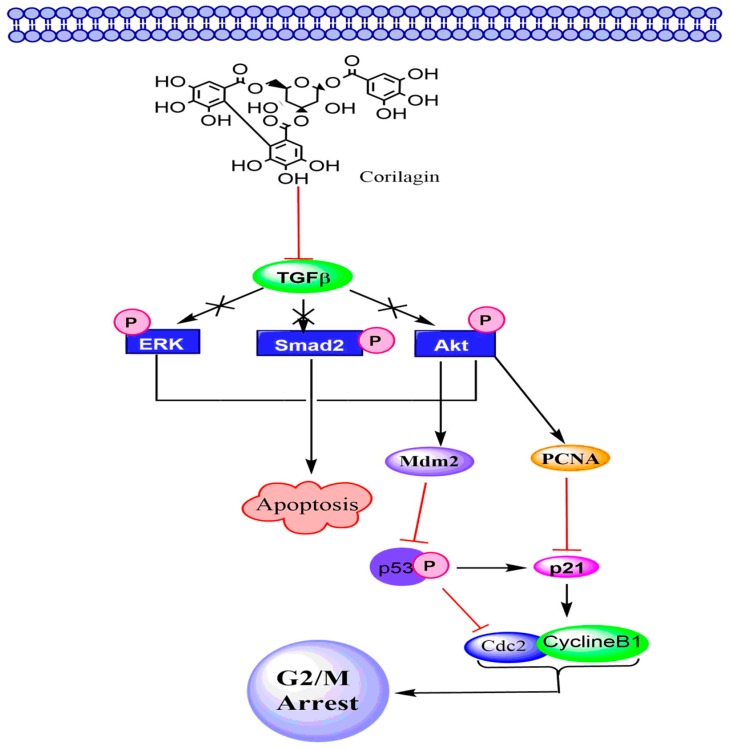
The effect of corilagin on the TGF-β signaling pathway in ovarian cancer.

**Table 1 molecules-24-03399-t001:** Distribution of corilagin in diverse plant groups.

Plant Part	Plant Species	Reference	Plant Part	Plant Species	Reference
Leaf	*Terminalia catappa* L.	[37]	Leaf	*Acalypha wilkesiana* Mull. Arg.	[46]
	*Acalypha hispida* Burm. f.	[46]		*Alchornea glandulosa* Poepp.	[47]
	*Jatropha curcas* L.	[48]		*Macaranga tanarius* Mull. Arg.	[49]
	*Mallotus japonicus* Mull. Arg.	[50]		*Phyllanthus muellerianus* (Kuntze) Exell.	[51]
	*Phyllanthus niruri* L.	[52]		*Phyllanthus urinaria* L.	[53]
	*Sapium insigne* (Royle) Benth.ex Hook.fil	[54]		*Phyllanthus amarus* Schumach. and Thonn.	[55]
	*Acer nikoense* (Miq.) Maxim.	[56]		*Acer amoenum* (Carriere)	[57]
	*Terminalia macroptera* Guill. and Perr.	[58]		*Lumnitzera racemosa* Willd.	[59]
	*Cunonia macrophylla* Brongn. and Gris	[60]		*Arctostaphylos uva-ursi* (L.) Spreng.	[61]
Fruit rind	*Dimocarpus longan* Lour.	[62]	Fruit rind	*Terminalia citrina* (Gaertn.) Roxb.	[63]
	*Nephelium lappaceum* L.	[64]		*Terminalia arjuna*	[65]
	*Terminalia. chebula* Retz.	[65]			
Seed, peel	*Euphorbia longana* Lam.	[66]	Flower	*Nymphaea stellata* Willd.	[67]
Aerial part	*Geranium sibiricum* L.	[37]	Aerial part	*Euphorbia pekinensis* Rupr.	[68]
	*Phyllanthus ussuriensis* Rupr. et Maxim.	[69]		*Geranium carolinianum* L.	[70]
	*Geranium pyrenaicum* Burm.f.	[71]		*Geranium potentillifolium* DC.	[72]
	*Geranium bellum* Rose	[72]		*Erodium stephanianum* Willd.	[73]
	*Pelargonium reniforme* Spreng.	[74]		*Erodium cicutarium* L’Hér. ex Aiton	[75]
	*Polygonum chinense* L.	[76]		*Saururus chinensis* (Lour.) Bail.	[77]
Whole plant	*Euphorbia prostrata* Aiton	[78]	Whole plant	*Excoecaria agallocha* L.	[79]
	*Cynanchum paniculatum* (bunge) kitagawa	[79]		*Phyllanthus tenellus* Roxb	[80]
	*Phyllanthus wightianus* Müll.Arg.	[81]		*Geranium thunbergii* Siebold ex Lindl. and Paxton	[82]
	*Caesalpinia coriaria* (Jacq.) Willd.	[83]		*Geranium wilfordii* Maxim	[84]
Fruit	*Phyllanthus emblica* L.	[34]		*Terminalia bellerica* (Gaertn.) Roxb.	[65]
	*Canarium album* L.	[85]		*Caraipa densifolia* Mart.	[86]
	*Zanthoxylum piperitum* (L.) DC.	[87]			

**Table 2 molecules-24-03399-t002:** In vitro anticancer effects of corilagin on various cancer cell lines.

Cancer Type	Cell Line	Effect	Mechanism	Reference
Breast cancer	MCF-7, SK-BR3	Apoptosis, autophagic cell death, necroptosis	↓Procaspase-3, ↓procaspase-8, ↓procaspase-9, ↓PARP, ↓Bcl-2, ↑caspase-8, ↑caspase-9, ↑Bax, ↑cleaved PARP	[28]
Cholangiocarcinoma	QBC9939, MZ-Cha-1	Antiproliferation, G_2_/M arrest	↓Bcl-2, ↑caspase-3, ↑p-Akt, ↑pErk1/2	[107]
Esophageal squamous cell carcinoma (ESCC)	ESCC cells	Antiproliferation, apoptosis	↑DNA damage, ↓DNA repair, ↓E3 ubiquitin ligase RNF8	[108]
Gastric carcinoma	SGC7901, BGC823	Apoptosis, antiproliferation	↑Caspase-3, ↑caspase-8, ↑caspase-9, ↑PARP	[114]
Glioblastoma multiforme	U251, T98G	Antiproliferation, apoptosis	↑Caspase-3, ↑caspase-7	[128]
Hepatocellular carcinoma	Bel7402, SMMC7721	Antiproliferation, G_2_/M arrest	↓p-Akt, ↓PCNA, ↑p-p53, ↑caspase-3, ↑caspase-9	[120,123]
Lung cancer	A549	Antiproliferation	↑ DNA damage	[30]
Ovarian cancer	A2780, SKOv3ip, Hey	Apoptosis, G_2_/M arrest	↓Cyclin B1, ↓Myt1, ↓phospho-cdc2, ↓phospho-Weel	[40]

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
