# Peer review of "Corilagin in Cancer: A Critical Evaluation of Anticancer Activities and Molecular Mechanisms"

_molecules, 2019, doi:10.3390/molecules24183399_

Round 1

Reviewer 1 Report

The manuscript is interesting and describes the actions of Corilagin in cancer.

I have only minor comments.

I suggest that the authors also read these recently manuscripts regarding the action of Corilagin:

Qiu F, Liu L, Lin Y, Yang Z, Qiu F. Corilagin Inhibits Esophageal Squamous

Cell Carcinoma by Inducing DNA Damage and Down-regulation of RNF8. Anticancer

Agents Med Chem. 2019 Mar 7.

Xu J, Zhang G, Tong Y, Yuan J, Li Y, Song G. Corilagin induces apoptosis,

autophagy and ROS generation in gastric cancer cells in vitro. Int J Mol Med.

2019 Feb;43(2):967-979.

 Yamada H, Wakamori S, Hirokane T, Ikeuchi K, Matsumoto S. Structural Revisions

in Natural Ellagitannins. Molecules. 2018 Jul 30;23(8).

Bai X, Pan R, Li M, Li X, Zhang H. HPLC Profile of Longan (cv. Shixia)

Pericarp-Sourced Phenolics and Their Antioxidant and Cytotoxic Effects.

Molecules. 2019 Feb 11;24(3). pii: E619.

 Tang YY, He XM, Sun J, Li CB, Li L, Sheng JF, Xin M, Li ZC, Zheng FJ, Liu GM,

Li JM, Ling DN. Polyphenols and Alkaloids in Byproducts of Longan Fruits

(Dimocarpus Longan Lour.) and Their Bioactivities. Molecules. 2019 Mar 26;24(6).

I suggest that the authors describe any adverse reactions of Corilagin

Author Response

General comment:

The manuscript is interesting and describes the actions of Corilagin in cancer.

I have only minor comments.

Response:

We greatly appreciate the encouraging comments regarding the quality of our work. As described below, we have revised our manuscript based on the reviewer’s constructive comments.

Comment 1:

I suggest that the authors also read these recently manuscripts regarding the action of Corilagin:

Qiu F, Liu L, Lin Y, Yang Z, Qiu F. Corilagin Inhibits Esophageal Squamous

Cell Carcinoma by Inducing DNA Damage and Down-regulation of RNF8. Anticancer

Agents Med Chem. 2019 Mar 7.

Xu J, Zhang G, Tong Y, Yuan J, Li Y, Song G. Corilagin induces apoptosis,

autophagy and ROS generation in gastric cancer cells in vitro. Int J Mol Med.

2019 Feb;43(2):967-979.

Yamada H, Wakamori S, Hirokane T, Ikeuchi K, Matsumoto S. Structural Revisions

in Natural Ellagitannins. Molecules. 2018 Jul 30;23(8).

Bai X, Pan R, Li M, Li X, Zhang H. HPLC Profile of Longan (cv. Shixia)

Pericarp-Sourced Phenolics and Their Antioxidant and Cytotoxic Effects.

Molecules. 2019 Feb 11;24(3). pii: E619.

Tang YY, He XM, Sun J, Li CB, Li L, Sheng JF, Xin M, Li ZC, Zheng FJ, Liu GM,

Li JM, Ling DN. Polyphenols and Alkaloids in Byproducts of Longan Fruits

(Dimocarpus Longan Lour.) and Their Bioactivities. Molecules. 2019 Mar 26;24(6).

Response:

We are thankful to the reviewer for suggesting these excellent papers. All the suggested publications have been cited at appropriate places in the manuscript (Yamada et al., 2018, reference no 16, page 2, lines 71-74; Tang et al., 2019, reference no. 21, page 2, lines 82 and 83; Bai et al., 2019, reference 30, page 2, line 84 and page 7, lines 259-264; Xu et al., 2019, reference114, page 6, line 225-235; Qiu et al., 2019, reference no.108, page 7, line 211-217).

Comment 2:

I suggest that the authors describe any adverse reactions of Corilagin.

Response:

Under section 7 (Safety Evaluation of Corilagin), several studies pertaining to safety aspects have been described. The studies indicated that corilagin was almost safe without any adverse effect and virtually nontoxic to normal cells or tissues (page 11, lines 397-409).

Reviewer 2 Report

The manuscript appears to be comprehensive in covering most of the literature with one exception, a recent paper (2019) by Qui et al. on the effects of Corilagin on esophageal squamous cell carcinoma (ESCC).  This is lacking in coverage and will need a paragraph.  

One other minor comment is that line 140 states "the anticancer potential of corilagin was started somewhere????"  Obviously this was a spacer left in the text.  I believe you are referring to Tanaka et al., 1985 from Kyushu University???

Author Response

Comment 1:

The manuscript appears to be comprehensive in covering most of the literature with one exception, a recent paper (2019) by Qui et al. on the effects of Corilagin on esophageal squamous cell carcinoma (ESCC).  This is lacking in coverage and will need a paragraph. 

Response:

This is a terrific suggestion! We have introduced a new section (5.3. Esophagel Cancer, page 6, lines 209-218) to include the study suggested.

Comment 2:

One other minor comment is that line 140 states "the anticancer potential of corilagin was started somewhere????"  Obviously this was a spacer left in the text.  I believe you are referring to Tanaka et al., 1985 from Kyushu University???

Response:

We thank the reviewer for the suggestion regarding incorporation of reference of Tanaka et al., 1985. We have revised the sentence and cited this important work (reference 88, page 5, line 150).

Reviewer 3 Report

Dear authors,

After the review process, I have several comments:

you should clearly present the aim of the paper in the abstract; you should delete section 2, it is not necessary because the paper is a review; you should improve section 4, it is poor represented; it represent a key factor to express the anticancer effects; you should develop conceptual frameworks to extend past researches; the authors presented only research insights, but existing gaps, and future research directions are poor detailed.

Best regards!

Author Response

Comments:

After the review process, I have several comments:

you should clearly present the aim of the paper in the abstract; you should delete section 2, it is not necessary because the paper is a review; you should improve section 4, it is poor represented; it represent a key factor to express the anticancer effects; you should develop conceptual frameworks to extend past researches; the authors presented only research insights, but existing gaps, and future research directions are poor detailed.

Best regards!

Response:

The aim of paper has been clearly presented in the abstract (page 1, lines 32 and 33) and introduction (page 2, lines 92-94).

Since we followed the Systematic Reviews and Meta-Analysis (PRISMA) criteria which are recommended for review articles by most of the reputed and high-impact biomedical science journals, we feel it is important to include this section. 

The “Bioavailability of Corilagin” section (section 4) has been modified and relevant information has been incorporated (page 3, lines 124-129 and page 3, line137 to page 4, line 144).

We have added a new section “Limitations and Future Prospects” (section 8, page 11, lines 411-422) to address the existing gap in knowledge, current limitation, and future research directions.

Round 2

Reviewer 3 Report

Dear Authors,

I do not have supplementary comments.

Best regards!